

# Supplementary feeding of cattle-yak in the cold season alters rumen microbes, volatile fatty acids, and expression of *SGLT1* in the rumen epithelium

Yuzhu Sha[1], Jiang Hu[1], Bingang Shi[1], Renqing Dingkao[2], Jiqing Wang[1], Shaobin Li[1], Wei Zhang[1], Yuzhu Luo[1] and Xiu Liu[1]

[1] College of Animal Science and Technology / Gansu Key Laboratory of Herbivorous Animal Biotechnology, Gansu Agricultural University, Lanzhou, Gansu, China
[2] Institute of Animal Husbandry Science of Gannan Prefecture, Hezuo, Gansu, China

Corresponding author
Xiu Liu, liuxiu@gsau.edu.cn

## ABSTRACT

Cattle-yak, a hybrid offspring of yak (*Bos grunniens*) and cattle (*Bos taurus*), inhabit the Qinghai-Tibet Plateau at an altitude of more than 3,000 m and obtain nutrients predominantly through grazing on natural pastures. Severe shortages of pasture in the cold season leads to reductions in the weight and disease resistance of grazing cattle-yak, which then affects their production performance. This study aimed to investigate the effect of supplementary feeding during the cold season on the rumen microbial community of cattle-yak. Six cattle-yak (bulls) were randomly divided into two groups—"grazing + supplementary feeding" (G+S) ($n = 3$) and grazing (G) ($n = 3$)—and rumen microbial community structure (based on 16S rRNA sequencing), volatile fatty acids (VFAs), and ruminal epithelial sodium ion-dependent glucose transporter 1 (*SGLT1*) expression were assessed. There were significant differences in the flora of the two groups at various taxonomic classification levels. For example, Bacteroidetes, Rikenellaceae, and *Rikenellaceae_RC9_gut_group* were significantly higher in the G+S group than in the G group ($P < 0.05$), while Firmicutes and *Christensenellaceae_R-7_group* were significantly lower in the G+S group than in the G group ($P < 0.05$). Kyoto Encyclopedia of Genes and Genomes (KEGG) and Clusters of Orthologous Groups (COG) analyses revealed that functions related to carbohydrate metabolism and energy production were significantly enriched in the G+S group ($P < 0.05$). In addition, the concentration of total VFAs, along with concentrations of acetate, propionate, and butyrate, were significantly higher in the G+S group than in the G group ($P < 0.05$). Furthermore, *SGLT1* expression in ruminal epithelial tissue was significantly lower in the G+S group ($P < 0.01$). Supplementary feeding of cattle-yak after grazing in the cold season altered the microbial community structure and VFA contents in the rumen of the animals, and decreased ruminal epithelial *SGLT1* expression. This indicated that supplementary feeding after grazing aids rumen function, improves adaptability of cattle-yak to the harsh environment of the Qinghai-Tibet Plateau, and enhances ability of the animals to overwinter.

## INTRODUCTION

The mammalian digestive system is colonized by complex microorganisms. In particular, the rumen contains numerous microorganisms, such as bacteria, fungi, archaea, and protozoa, which participate in plant cell-wall hydrolysis and digestion of fibrous materials in the rumen. These materials are converted into absorbable compounds, such as bacterial proteins and volatile fatty acids (VFAs) (*Nagaraja, 2016*; *Yu et al., 2020*). Among them, VFAs are the main energy source of ruminant hosts, providing 70–80% of the energy requirements (*Russell & Rychlik, 2001*). VFAs are predominantly absorbed through the rumen epithelium to the blood via VFA transporters (*Yohe et al., 2019*). The rumen epithelium, a unique niche for interactions between the host and microorganisms, not only represents a physical barrier to the contents of the lumen but also affects net utilization of nutrients throughout the body (*Lin et al., 2019*; *Malmuthuge & Guan, 2017*). Glucose in the rumen is decomposed into VFAs by commensal microorganisms, but some glucose is transported directly to the blood via sodium ion-dependent glucose transporter 1 (*SGLT1*) (*Aschenbach et al., 2000a*). In this way, the energy consumed in the rumen fermentation process is saved and the glucose is directly transported to the blood to supply energy for the body (*Aschenbach et al., 2000b*; *Aschenbach, Borau & Gabel, 2002*). Many studies have demonstrated a symbiotic relationship between intestinal microflora and the host. For example, changes in diet and physiology of the host can result in changes in gut microbes (*Sun et al., 2016*).

The cattle-yak is a hybrid offspring of yak (*Bos grunniens*) and cattle (*Bos taurus*), and has obvious heterosis (*Wiener, Han & Long, 2011*). The cattle-yak provides herders with meat, fur, fuel (animal dung as fuel), and other factors, and is an important component of animal husbandry in the Qinghai-Tibet Plateau. Like the yak, the cattle-yak lives at high-altitude and mainly obtains nutrients by grazing on natural pastures. The unique geographic environment of the alpine steppe area of the Qinghai-Tibet Plateau means each year is divided into a cold season and a warm season, which correspond to the grass period and hay period (when the grassland vegetation changes), respectively (*Liu et al., 2020*). In the hay period, grazing livestock such as yak and cattle-yak can only obtain nutrients by consuming hay, which seriously affects their production performance and results in various problems such as weight loss and lack of disease resistance (*Sun, Angerer & Hou, 2015*; *Xin et al., 2011*). Supplementary feeding of high-altitude mammals after grazing in the cold season can improve the physical condition, immunity, and fertility of the animals, allowing them to successfully survive winter (*Jing et al., 2018*; *Jing et al., 2017*). However, there are limited reports on changes in the rumen microbial community, rumen VFAs, and related gene expression in the rumen tissues of the plateau cattle-yak after supplementary feeding combined with grazing in the cold season. We hypothesized that after supplemental feeding of cattle-yak in the harsh environment of the cold season, the microbial community, fermentation product VFAs, and gene expression in the rumen epithelium would undergo a series of changes. This study aimed to test this hypothesis by comparing rumen microorganisms, VFAs, and *SGLT1* gene expression in cattle-yak
that grazed only (G) with those that were also fed supplementary feed (G+S) in the cold season.

## MATERIALS & METHODS

### Ethics approval and sampling

All experimental protocols were approved by the Livestock Care Committee of Gansu Agricultural University (GAU-LC-2020-055), and the experiment was conducted with the consent of the herdsman (Jie Gazang). Six cattle-yak (bulls) were selected from the cattle-yak (Gannan yak ♀ × Jersey cattle ♂) of a single herder in Gannan Tibetan Autonomous Prefecture, Gansu Province (359 ± 3.58 kg; 2-3 years old). In the same cowshed, the animals were randomly divided into a "grazing + supplementary feeding" group (G+S; $n = 3$) and a grazing control group (G; $n = 3$), separated by fences. The two groups grazed in the same pasture during the day. At 6 PM, cattle-yaks of the G+S group were given access to supplementary feed of grass (Highland barley grass and Oat grass) + highland barley (concentrate) in a ratio of 3:7 concentrate to forage. Supplementary feeding was for a period of five months (from November to March of the following year). After the supplementary feeding period ended, the cattle-yak were slaughtered. Rumen organs were removed immediately after slaughter and contents of the rumen abdominal sac were collected. Fifty milliliters of rumen contents were obtained from each cattle-yak, filtered through four layers of sterile gauze, divided into three sterile cryogenic tubes, and immediately stored in liquid nitrogen for subsequent 16S rRNA gene analysis and determination of VFA concentrations. In addition, rumen epithelial tissue samples were acquired by dissecting a small piece of rumen abdominal sac, quickly removing rumen contents by rinsing with phosphate-buffered saline (PBS), then separating the epithelial tissue with blunt scissors. The tissue was immediately placed in liquid nitrogen and stored for subsequent RNA extraction to assess *SGLT1* expression.

### DNA extraction and high-throughput sequencing

A MN NucleoSpin 96 Soil kit (Macherey-Nagel, Germany) was used to extract microbial DNA from the rumen content samples, and the DNA concentration and purity were detected by NanoPhotometer (N60, Germany). All DNA samples were stored at −80 °C until subsequent processing. For analysis of the community structure of the rumen microorganisms, theV3-V4 region of the 16S rRNA gene was amplified from the DNA sample by PCR using universal primers 338F (5′-ACTCCTACGGGAGGCAGCAG-3′) and 806R (5′- GGACTACHVGGGTWTCTAAT-3′). The PCR comprised a pre-denaturation step at 95 °C for 3 min, 40 cycles of denaturation at 95 °C for 30 s, annealing at 55 °C for 30 s, and extension at 72 °C for 30 s, followed by a final extension at 72 °C for 7 min. The library obtained by this PCR amplification process was sequenced on an Illumina MiSeq platform (Illumina, San Diego, CA, USA), and bioinformatics analysis was performed using BMKCloud (http://www.biocloud.net).

### Determination of rumen VFAs

Concentrations of VFAs were determined using a gas chromatograph (GC-2010 Plus; Shimadzu, Kyoto, Japan) with an AT-free fatty acid phase (FFAP) chromatographic

**Table 1 Primer sequences for mRNA expression analysis.**

| Gene | Primer sequence (5′–3′) | Length (bp) | Temperature (°C) | GenBank accession number |
|------|-------------------------|-------------|------------------|--------------------------|
| *SGLT1* | F: GTTTGCCTATGGAACCGGGA | 148 | 60 | NM_174606.2 |
| | R: TGCAATGGGCTTGGTGAAGA | | | |
| *β-actin* | F: CGCAAGTACTCCGTGTGGAT | 146 | 60 | NM_173979.3 |
| | R: TAACGCAGCTAACAGTCCGC | | | |

**Notes.**
SGLT1, sodium ion-dependent glucose transporter 1.

column (30 m × 0.32 mm × 0.25 m). The internal standard method was employed using 2-ethyl butyric acid (2EB) as the internal standard. The detector temperature of the chromatograph was 260 °C, the temperature of the injection port was 250 °C, and the chromatographic column heating procedure comprised maintenance of temperature at 60 °C for 1 min, increase to 115 °C at 5 °C/min (without maintenance), and then increase to 180 °C at 15 °C/min.

## mRNA expression analysis

RNA isolater reagent (Vazyme, Nanjing, China, 7E371D9) was used to extract RNA from ruminal epithelial tissues. Concentration and purity of the extracted RNA were measured using an ultra-micro spectrophotometer (NanoDrop 2000; Thermo Scientific). cDNA synthesis was performed using a reverse transcription kit containing gDNA wiper (R323-01). Primer 5.0 software was used to design the gene primers (Table 1). A Q6 real-time fluorescence quantitative PCR system (Applied Biosystems) was used to quantify expression of *SGLT1* and the internal reference gene (*β*-actin). The reaction conditions comprised a pre-denaturation step at 95 °C for 30 s, followed by 40 cycles of 95 °C for 10 s and 60 °C for 30 s. The dissolution curve analysis conditions were 95 °C for 15 s, 60 °C for 60 s, and 95 °C for 15 s. The 20-μl reaction system contained 2×ChamQ Universal SYBR qPCR Master Mix (Q711-02), the cDNA template, and the upstream and downstream primers. The data analysis was performed using the $2^{-\Delta\Delta CT}$ method (*Livak & Schmittgen, 2001*).

## Bioinformatics analysis

Raw data returned by the Illumina MiSeq platform were subjected to merging of paired-end reads (FLASH v1.2.7), filtering (Trimmomatic v0.33), and removal of chimeras (UCHIME v4.2) to obtain optimized sequences (tags). UCLUST in QIIME (version 1.8.0) software was used to cluster tags and obtain operational taxonomic units (OTUs) at the 97% similarity level. OTUs were annotated based on the SILVA bacterial taxonomy database. Taxonomic analysis was subsequently performed at different taxonomic levels (phylum, class, order, family, and genus) to obtain community structure maps and species clustering heatmaps. Species diversity within taxa was analyzed using the $α$-diversity indices ACE, Chao1, Shannon, and Simpson, and rarefaction curves were plotted (*Wang et al., 2012*). $β$-diversity analysis was employed to compare differences in species diversity (microbial

**Table 2  Effect of supplementary feeding after grazing on diversity index.**

| Group | Number of OTUs | ACE | Chao1 | Simpson | Shannon | Coverage |
|-------|----------------|--------|--------|---------|---------|----------|
| G+S | 498 | 501.85 | 505.06 | 0.04 | 4.33 | 0.10 |
| G | 499 | 500.16 | 500.07 | 0.05 | 4.49 | 0.10 |
| *P* value | 0.53 | 0.48 | 0.69 | 0.51 | 0.51 | 0.69 |

**Notes.**

G, grazing; G+S, grazing + supplementary feeding.

Chao1 and Ace indices measure species abundance; Shannon and Simpson indices measure species diversity; Coverage index represents OTU coverage.

composition and community structure) between groups. PICRUSt software was used to predict metagenomic functional profiles of the microbial communities in each group (identified by 16S rRNA gene sequencing) based on both the Kyoto Encyclopedia of Genes and Genomes (KEGG) and Clusters of Orthologous Groups (COG) databases.

## Statistical analysis

Significance analysis of between-group differences (linear discriminant analysis effect size (LefSe) analysis) was used to explore potential biomarkers with statistical differences between the two groups (*Segata et al., 2011*). Metastats software was used to perform *t*-tests on the between-group differences in species relative abundance (*White, Nagarajan & Pop, 2009*) and obtain *P* values, which were corrected to obtain q values. Species that exhibited significant differences in relative abundance between the two groups (according to the q values) could then be identified. SPSS software v24.0 (SPSS, Inc., Chicago, Illinois) was used to analyze the data. Independent-sample *t*-tests were used to analyze the differences in rumen VFAs and *SGLT1* expression between the G+S and G groups in the cold seasons. Spearman correlation analysis was used to analyze the correlation between VFA levels and *SGLT1* expression. *P* values of 0.05 (two-tailed) and 0.01 (two-tailed) indicated significant differences, respectively.

## RESULTS

### 16S rRNA sequencing and rumen bacterial diversity

A total of 160,213 pairs of reads were obtained by 16S rRNA gene sequencing (paired-end), and 127,677 clean tags were generated after merging of paired-end reads and filtering. Each sample generated at least 63,772 clean tags and a mean of 63,839 clean tags (with a mean sequence length of 412 bp). Tags were clustered at a similarity level of 97% using USEARCH software and 502 OTUs were obtained. There were 498 OTUs in the G+S group and 499 in the G group, with 3 and 4 unique OTUs in the G+S and G groups, respectively (Fig. S1A). Rarefaction curves (Fig. S1B) indicate the species diversity and species richness in both groups. Limited new OTUs were observed after approximately 20,000 reads, indicating that the sequencing coverage was sufficient. There were no significant differences in the $\alpha$-diversity indices between the two groups ($P > 0.05$) (Table 2).
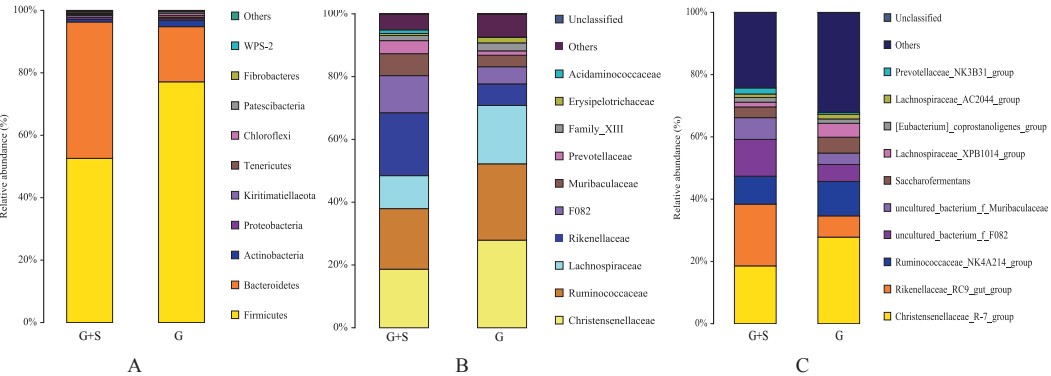

**Figure 1  Rumenmicrobial species composition of cattle-yak in grazing group and supplementary feeding group under different taxonomic classification levels.** (A) Phylum; (B) Family; (C) Genus. Each bar represents the average relative abundance of each bacterial taxon within a group. G: grazing; G + S: grazing + supplementary feeding.

## Microbial community structure in the rumen

Eleven phyla were detected (Data S1), with Firmicutes, Bacteroidetes, and Actinobacteria dominating both groups. The sum of the relative abundances of Firmicutes and Bacteroidetes accounted for >94% of all taxa in both groups. The relative abundance of Firmicutes was significantly lower in the G+S group than in the G group ($P < 0.05$), but the relative abundance of Bacteroidetes was significantly higher ($P < 0.05$) (Fig. 1A). In addition, the Firmicutes/Bacteroidetes ratio was significantly lower in the G+S group (1.2035) than in the G group (4.3650; $P < 0.05$). At the family level (Fig. 1B), Christensenellaceae and Ruminococcaceae were the dominant families, and the relative abundance of Rikenellaceae was significantly higher in the G+S group than in the G group ($P < 0.05$). At the genus level (Fig. 1C), 123 bacterial taxa were detected (Date S1), with *Christensenellaceae_R-7_group*, *Rikenellaceae_RC9_gut_group*, and *Ruminococcaceae_NK4A214_group* being dominant. Among them, the relative abundance of *Christensenellaceae_R-7_group* was significantly lower in the G+S group than in the G group ($P < 0.05$), while the relative abundance of *Rikenellaceae_RC9_gut_group* was significantly higher ($P < 0.05$).

## Metagenomic functional profile predictions of the microbial communities

PICRUSt software predicted the gene families related to the microbes identified by 16S rRNA gene sequencing, and 43 KEGG gene families and 25 COG gene families were identified (Data S2). Among the 43 KEGG gene families, >60% were associated with metabolism, with the largest proportion being related to carbohydrate metabolism. The function of carbohydrate metabolism was significantly less common in the G+S group than in the G group ($P < 0.05$) (Fig. 2). However, the function of glycan biosynthesis and metabolism was significantly more common in the G+S group than in the G group. Second, Global and Overview Maps, Amino acid metabolism and Energy metabolism accounted for the largest proportion, and the functions of Amino acid metabolism and

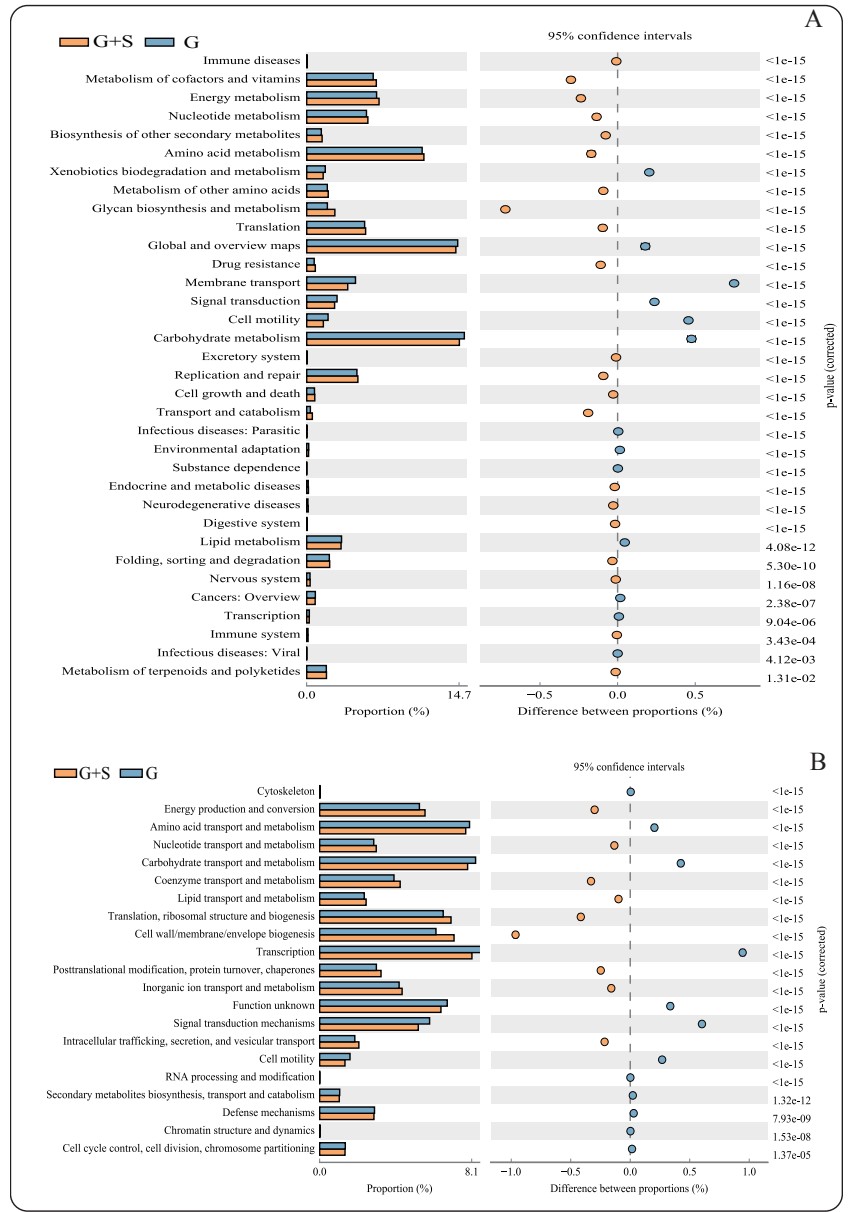

**Figure 2 PICRUSt functional predictions based on the Kyoto Encyclopedia of Genes and Genomes (KEGG) and Clusters of Orthologous Groups (COG) databases.** (A) KEGG database; (B) COG databases. Bars on the left indicate the abundance ratio for each function in the two sets of samples; the graph on the right shows the percentage of difference in functional abundance within the 95% confidence interval; and *P* values are displayed on the far right of each panel. G: grazing; G + S: grazing + supplementary feeding.

Energy metabolism were significantly higher in the G+S group than in the G group. Among the 25 COG gene families, transcription was the most frequent. In addition, the most abundant functional categories of Carbohydrate transport and metabolism, Amino acid transport and metabolism, and Energy production and conversion were significantly different between the two groups of cattle-yak. Carbohydrate transport and Amino acid

**Table 3  Effects of supplementary feeding after grazing on rumen fermentation parameters and *SGLT1* gene expression.**

| Ruminal VFAs (mmol/L) | G+S | G | *P* |
|---|---|---|---|
| Acetate | 47.97 ± 3.26 | 36.75 ± 0.39 | 0.004 |
| Propionate | 17.94 ± 0.71 | 15.47 ± 0.09 | 0.004 |
| Butyrate | 7.30 ± 0.00 | 6.46 ± 0.10 | 0.000 |
| Isobutyrate | 1.41 ± 0.03 | 1.32 ± 0.02 | 0.011 |
| Isovalerate | 2.44 ± 0.14 | 2.53 ± 0.07 | 0.406 |
| Valerate | 1.47 ± 0.11 | 1.29 ± 0.04 | 0.064 |
| Total VFA | 78.54 ± 3.68 | 63.82 ± 0.71 | 0.002 |
| Acetate/propionate ratio | 2.67 ± 0.08 | 2.38 ± 0.01 | 0.003 |
| **Gene expression** | | | |
| *SGLT1* | 1.08 ± 0.10 | 5.56 ± 0.61 | 0.005 |

**Notes.**
G, grazing; G+S, grazing + supplementary feeding; SGLT1, sodium ion-dependent glucose transporter 1; VFA, volatile fatty acid.

transport and metabolism were significantly higher in the G group than in the G+S group. However, Energy production and conversion was significantly lower in the G group than in the G+S group, which is consistent with the KEGG functional analysis.

## Rumen VFAs and ruminal epithelial *SGLT1* expression

The total concentration of VFAs was significantly higher in the G+S group than in the G group ($P < 0.01$). Acetate, propionate, butyrate, and isobutyrate concentrations were also significantly higher in the G+S group than in the G group ($P < 0.01$), but there were no significant differences between the two groups in the other VFAs (valerate and isovalerate) ($P > 0.05$). Furthermore, the acetate/propionate ratio was significantly higher in the G+S group than in the G group ($P < 0.01$) (Table 3). Ruminal epithelial *SGLT1* expression was significantly lower in the G+S group than in the G group ($P < 0.01$). Correlation analysis revealed that VFAs had a definite association with the *SGLT1* gene; isobutyrate, butyrate, and valerate were significantly negatively correlated with *SGLT1* expression ($P < 0.05$), while isovalerate had a low correlation with *SGLT1* expression (Fig. 3).

## DISCUSSION

The rumen microbial community structure of ruminants is affected by numerous factors such as feed variety, diet composition, seasonal changes, and feeding management (*Yamano et al., 2019*; *Li et al., 2019*). The current study analyzed microbial communities in the rumen of cattle-yak using 16S rRNA gene sequencing and revealed that supplementary feeding after grazing had little effect on the $\alpha$-diversity index of microorganisms. This indicates that supplementary feeding after grazing did not change the microbial community diversity of the rumen, even though there were changes in the community composition and relative abundances of specific microorganisms. At the phylum level, Firmicutes, Bacteroidetes, and Actinobacteria were dominant in both groups of cattle-yak. Firmicutes possess many genes encoding energy metabolism-related enzymes that break down various substances

|          | Acetate | Propionate | Isobutyrate | Butyrate | Isovalerate | Valerate | SGLT1 |
|----------|---------|------------|-------------|----------|-------------|----------|-------|
| Acetate | 1.000 | 1.000** | 0.771 | 0.771 | - 0.486 | 0.771 | - 0.771 |
| Propionate | 1.000** | 1.000 | 0.771 | 0.771 | - 0.486 | 0.771 | - 0.771 |
| Isobutyrate | 0.771 | 0.771 | 1.000 | 1.000** | - 0.029 | 1.000** | - 0.886** |
| Butyrate | 0.771 | 0.771 | 1.000** | 1.000 | - 0.029 | 1.000** | - 0.886** |
| Isovalerate | - 0.486 | - 0.486 | - 0.029 | - 0.029 | 1.000 | - 0.029 | 0.257 |
| Valerate | 0.771 | 0.771 | 1.000** | 1.000** | - 0.029 | 1.000 | - 0.886** |
| SGLT1 | - 0.771 | - 0.771 | - 0.886** | - 0.886** | 0.257 | - 0.886** | 1.000 |

-1          0          1

**Figure 3  Correlation matrix between VFAs and *SGLT1* gene.** Asterisks indicate significance of correlation at the 0.05 (*) and 0.01 (**) levels (both two-tailed). SGLT1: sodium ion-dependent glucose transporter 1.

and aid host digestion and absorption of nutrients (*Kaakoush, 2015*). Bacteroidetes can degrade carbohydrates and proteins (*Nuriel-Ohayon, Neuman & Koren, 2016*; *Fernando et al., 2010*; *Jami, White & Mizrahi, 2014*), and a high Firmicutes/Bacteroides ratio can help hosts effectively absorb energy-related nutrients and maintain their metabolic balance in low-temperature environments (*Ley et al., 2006*; *Fernando et al., 2010*; *Murphy et al., 2010*). In the current study, cattle-yak in the G+S group were affected by dietary composition (mainly roughage, supplemented with non-cellulose concentrate), which caused an increase in the content of Bacteroidetes and the ratio of Firmicutes/Bacteroides (*Fernando et al., 2010*), and consequently the G+S cattle-yak could absorb energy-related nutrients more effectively and maintain their metabolic balance in the low-temperature environment (*Murphy et al., 2010*; *Jami, White & Mizrahi, 2014*).

At the family level, there were some differences in the microbiota between the two groups of cattle-yak. Christensenellaceae and Ruminococcaceae, which are beneficial flora, were the predominant bacterial families in this study. *Waters & Ley (2019)* found that Christensenellaceae is an important family of the phylum Firmicutes and is instrumental to human health. Rikenellaceae, a crucial polysaccharide-degrading family, displayed the highest abundance in the G+S group. This may be related to the supplement feeding of concentrate, and by degrading more polysaccharides, extra energy is generated for the host (*Laursen et al., 2017*). At the genus level, *Christensenellaceae_R-7_group* belongs to the phylum Firmicutes (*Waters & Ley, 2019*) and mainly decomposes fibrous substances (*Evans et al., 2011*). Cattle-yak in the G group mainly fed on dry grass (high fiber), which may account for the highest relative abundance of Firmicutes and *Christensenellaceae_R-7_group* in this group. *Rikenellaceae_RC9_gut_group* is involved in degrading plant-derived polysaccharides (*Seshadri et al., 2018*), and its relative abundance increases with increased dietary fiber content (*Qiu et al., 2019*). In the current study, the relative abundance of *Rikenellaceae_RC9_gut* likely increased in the G+S group because cattle-yak in this group consumed more concentrate (mainly roughage with non-cellulose concentrate added). Some studies have reported that *Rikenellaceae_RC9_gut_group* is related to the primary or secondary degradation of carbohydrates (*Pitta et al., 2010*; *Ramos et al., 2018*). Therefore,

appropriate supplementary feeding after grazing in the cold season is speculated to promote degradation of carbohydrates in the feed, which allow the cattle-yak to cope more effectively with the harsh cold season.

PICRUSt software was used to predict microbial gene family functions. Cattle-yaks in G group consumed some dried herbage with high fiber content, thus these animals need a large quantity of microorganisms to degrade the carbohydrates with high cellulose content (*Sun, Angerer & Hou, 2015*), and this leads to enrichment of carbohydrate metabolism. In the KEGG analysis, the function of glycan biosynthesis and metabolism was enriched in the G+S group. This was most likely because cattle-yaks in the G+S group consumed concentrate, which allows increased sugar production. In addition, energy-related metabolic enrichment was found in both KEGG and COG functional databases due to the addition of certain concentrates to meet specific energy requirements in the G+S group. Concurrently, amino acid metabolism was enriched in the G+S group. This suggests that after supplementation with certain concentrates, amino acid metabolism might also provide some energy for the host.

Research has demonstrated that >70% of VFAs are absorbed by the rumen epithelium, and these are the main energy source for ruminants (*Malmuthuge, Griebel & Guan, 2015*; *Russell & Rychlik, 2001*). In this study, the G+S group of cattle-yaks had the highest concentrations of total VFA, acetate, propionate, and butyrate. Supplementary feeding with an appropriate amount of concentrate (which increases the soluble carbohydrate content of the diet) was previously shown to lead to more ruminal VFA production by microbial fermentation, thereby providing energy to the host (*Qiu et al., 2019*). In addition, a diet focused on cellulose reduced the total production of VFAs in the rumen, whereas feeding a certain amount of non-fibrous material increased propionate production (*Polyorach, Wanapat & Cherdthong, 2014*; *Liu et al., 2019*). These findings are consistent with the results of the current study. The diet of cattle-yak in the G+S group was supplemented with non-cellulose concentrate, resulting in increased propionate content, while cattle-yak in the G group had long-term intake of hay, which decreased the total VFA production. Furthermore, the acetate/propionate ratio is correlated with feed energy use efficiency (*Baldwin, 1998*). This ratio was lower in cattle-yaks of group G than those of group G+S because group G had long-term intake of forage with high fiber content, which enabled the cattle-yaks in this group to use energy more efficiently.

Nutrients are not only absorbed via the rumen epithelium in the form of VFAs; some glucose is directly absorbed by *SGLT1* (*Aschenbach, Borau & Gabel, 2002*; *Aschenbach et al., 2000a*). The relative energy associated with glucose rather than VFAs is much higher in ruminants (which rely almost entirely on gluconeogenesis to meet their glucose needs) than in monogastric species (*Reynolds, Harmon & Cecava, 1994*). Direct absorption of glucose via *SGLT1* reduces the metabolic expenditure associated with gluconeogenesis in animals (*Aschenbach et al., 2000b*). In this study, expression of *SGLT1* was lower in the G+S group than in the G group. A possible explanation is that to meet the energy needs of the cattle-yak in the G group, nutrients were absorbed in the form of VFAs and glucose was also directly absorbed via SGLT1 (*Aschenbach et al., 2000a*); this would aid the cattle-yak in coping with the harsh cold season. Finally, correlation analysis demonstrated a negative

association of *SGLT1* expression with VFAs. This indicated that ruminants absorb energy in the form of VFAs but can also absorb glucose via *SGLT1* to obtain energy. When VFA concentrations decrease and energy supply is insufficient, *SGLT1* gene expression increases and energy is obtained directly by absorbing glucose, thus saving the energy required for fermentation.

## CONCLUSIONS

Intestinal microorganisms are an intrinsic part of the host lifecycle and influence animal phenotypes. Supplementary feeding in the cold season can improve the welfare of cattle-yak, which enhances production performance, disease resistance, and wintering ability. In this study, the rumen microbial flora of cattle-yaks exposed to two different grazing regimes were determined, and the interaction between cattle-yak rumen microbes and their metabolites and nutrient absorption-related genes of the host and their regulatory mechanisms were explored. Supplementary feeding in the cold season significantly increased the abundance of microbial flora and the VFA contents in the rumen of cattle-yak. There were also significant differences in the expression of *SGLT1* in rumen epithelial tissue. Microbial function prediction revealed that carbohydrate metabolism and energy production functions were significantly enriched following supplementary feeding, and this interaction may play a specific regulatory role in the process of energy production. Therefore, supplementary feeding after returning to grazing in the cold season can help cattle-yaks better adapt to the harsh environment in the Qinghai-Tibet Plateau, which improves their wintering ability and welfare. Data from this study also provide a basis for future research on cattle-yak cold adaptation evolution in the Qinghai-Tibet Plateau.

### Funding
This work was supported by the Basic Research Innovation Group Program of Gansu Province (17JR5RA137). The funders had no role in study design, data collection and analysis, decision to publish, or preparation of the manuscript.

### Grant Disclosures
The following grant information was disclosed by the authors:
Basic Research Innovation Group Program of Gansu Province: 17JR5RA137.

### Competing Interests
The authors declare there are no competing interests.

### Author Contributions

- Yuzhu Sha conceived and designed the experiments, performed the experiments, analyzed the data, prepared figures and/or tables, and approved the final draft.
- Jiang Hu and Yuzhu Luo conceived and designed the experiments, authored or reviewed drafts of the paper, and approved the final draft.

- Bingang Shi, Jiqing Wang and Shaobin Li performed the experiments, authored or reviewed drafts of the paper, and approved the final draft.
- Renqing Dingkao performed the experiments, authored or reviewed drafts of the paper, acquisition of experimental materials, and approved the final draft.
- Wei Zhang performed the experiments, analyzed the data, prepared figures and/or tables, acquisition of experimental materials, and approved the final draft.
- Xiu Liu conceived and designed the experiments, analyzed the data, prepared figures and/or tables, authored or reviewed drafts of the paper, and approved the final draft.

## Animal Ethics

The following information was supplied relating to ethical approvals (i.e., approving body and any reference numbers):

Livestock Care Committee of Gansu Agricultural University provided full approval for this research (GAU-LC-2020-055).

## Ethics

The following information was supplied relating to ethical approvals (i.e., approving body and any reference numbers):

All the experimental protocols were approved by the Livestock Care Committee of Gansu Agricultural University (GAU-LC-2020-055).

## Field Study Permissions

The following information was supplied relating to field study approvals (i.e., approving body and any reference numbers):

Field experiments were approved by the Livestock Care Committee of Gansu Agricultural University (project number:GAU-LC-2020-055); Experiment was conducted with the consent of the herdsman (Jie Gazang).

## DNA Deposition

The following information was supplied regarding the deposition of DNA sequences:

The data are available in the Sequence Read Archive (SRA): SRR12695872; SRR12695873; PRJNA662399; SAMN16083866.

## Data Availability

Raw data is available in the Supplementary Files.

## Supplemental Information

Supplemental information for this article can be found online at http://dx.doi.org/10.7717/peerj.11048#supplemental-information.

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
