# Peer review of "Supplementary feeding of cattle-yak in the cold season alters rumen microbes, volatile fatty acids, and expression of SGLT1 in the rumen epithelium"

_PeerJ, doi:10.7717/peerj.11048_

## Round 0.1 · original submission · Major Revisions

Overall, the topic of the manuscript is of interest. However, many points are needed to be clear before making a decision. Please revise this manuscript according to all reviewer's comments, point by point.

Reviewer 1 ·

Basic reporting

The manuscript describes changes in the microbial composition of rumen samples in two groups of cattle-yak receiving different diets. The study provides an interesting point of view to evaluate the benefits of supplementing feed with concentrates, in an effort to increase animal welfare and resistance to harsh conditions. This work poses some interesting basis for further research, given the hints to the socio-cultural background that would potentially benefit from such a study.
The work seems well organized, although improvements could be done throughout the whole manuscript.
The introduction section presents well the context and the general purpose of the study, but a clear hypothesis is missing. I understand the work is an exploratory study, but predictions and expectations could be formulated in this section, discussed and confirmed or rejected in the conclusion part. The conclusions section itself is more a repetition of results than a real conclusive statement. I would recommend rewriting this section and mentioning the limitations of the study.
I could not find a reference for the deposited raw sequences.
I noticed that many of the references cited are quite old (before 2010). I know many papers, especially regarding ruminant physiology, are quite dated, but I think there is a lot of recent material that could be consulted, for example, on microbial activity and pathways. I would suggest trying to update at least some of the references.
Tables’ headings could be expanded, adding descriptions that would help the reader to better understand the contents without the need to look for the references in the text. This could also be done for figures. Please include acronyms explanations in figures/tables descriptions.
Please check though all the manuscript: all bacterial taxa (phyla, genera, species, etc) must be formatted in italics.
Furthermore, I would suggest a native speaker to check the English language throughout the manuscript.
Please see the general comments section for specific comments per each line.

Experimental design

Methods description could be improved, as some sections seem lacking important details and the experimental planning could be clarified.
It would be crucial to add details about the animal husbandry. Can the authors provide information about the animals housing conditions? How were the groups kept separated? How did they receive the supplemental feeding?
Details regarding the sampling procedure are also missing. How was the rumen fluid collected and how much? Which region of the organ was sampled? Which instruments were employed and how was cross-contamination avoided? Additionally, may I ask to which purpose were the animals slaughtered? Was it only for the experiment?
Please add specific details regarding the kits used for DNA extraction and how were concentration and purity checked. How was the library prepared? Please provide more details on the sequencing process.
The authors did not specify why they choose to investigate only SGLT1. It seems odd to test the expression of only one gene: weren’t other genes investigated? Was β-actin the only housekeeping gene tested?
It is not clear from the text how were bioinformatical analyses performed. I would suggest to clearly state what software was used for each of the steps.
I would strongly recommend including plots showing beta-diversity. Why was beta-diversity not discussed at all?

Validity of the findings

The total number of reads and consequently of OTUs is very low. Do the authors have an explanation for that?
The authors detected several KEGG pathways that could be discussed. Nevertheless, the discussion focuses only on carbohydrate metabolism, with unexpected findings, I would say. The G+S group was receiving a carbohydrate supplement, therefore it would be expected to find higher activity associated with carbohydrate metabolism in this group. How do the authors explain the higher carbohydrate metabolism-related activity in the group that did not receive the supplement? I would recommend discussing also the other outcomes of the PICRUSt analysis.
Could the higher expression of SGLT1 in the G group be due to any other reasons, apart the ones discussed? Could the higher activity of microbes in G+S group ferment most of the glucose available and therefore cause a relative lower expression of the SGLT1 gene?
As stated previously, the conclusions seem a repetition of the results section. I would strongly recommend including in the discussion and in the conclusive statement the limitations of the study. I might suggest taking into account, for example, the limited number of samples and the possibility to induce ruminal acidosis with a high content of concentrate in the diet. Furthermore, include a confirmation/rejection of the hypothesis. Do the results confirm the hypothesis? Are the expectations on the outcome of the experiment met?

Additional comments

Please consider the following comments for specific sections of the manuscript:

Line 22: areas
Lines 25-26: use n=3 in brackets after each group description, instead of writing “three each”
Line 40: please rephrase “after grazing”
Line 41: please specify the geographical location
Line 45: delete “of ruminants”, as it sounds repetitive
Line 50: is there a reference for this statement?
Line 57: Aschenbach et al. 2000 is cited twice
Line 59: please consider rephrasing
Lines 62-63: could the authors specify more clearly how are the animals used for “hide and fuel”?
Lines 64-70: please add references for these statements
Line 73: supplementing feed in winter only provides positive effects?
Line 85: please consider adding the age of the animals
Line 86: please consider rewriting (see comment for lines 25-26)
Line 97: what is the name of the kit?
Line 98: how were concentration and quality assessed?
Line 132: how was the taxonomy assigned? Which version of the database was used?
Lines 139-140: please consider rephrasing
Lines 143-146: this part should probably be included in the statistical analyses part
Line 155: how were values equal to 0.05 considered?
Line 159: what to the authors mean with pairs of reads? Weren’t the reads merged as a first step of bioinformatical analyses?
Line 160: what do the authors mean with “clean” tags?
Lines 164-166: the rarefactions curves in the figure only show the number of OTUs. Please consider rephrasing.
Line 169: plural of phylum is phyla
Line 196: please specify “other VFAs”
Line 205: please consider rephrasing; for animals it would be more appropriate to use “feed” instead of “food”
Lines 235-238: if I understand correctly, the authors suggest that G+S group eat more fiber with their diet. How is that possible if they were receiving grain as supplementation?
Lines 252-255: Please add references.
Lines 263-264: what do the authors mean with “long-term intake of hay”? When did the animals receive hay?
Lines 275-281: How can the authors state that when VFAs decrease SLGT1 expression increases? Correlation indicates only the degree of relationship between two variables, please consider a regression analysis to strengthen this point, or add a reference to previous studies.
Line 284: please rephrase “rumen internal environment”
Lines 286-294: do not repeat all the results in the conclusion, but just focus on the most relevant.

Table 2: could the authors explain the indexes in the description?
Table 3: please report the values also for each of the other VFAs measured.
Figure 1: may I suggest trying colorblind-friendly palettes for the bar charts?

·

Basic reporting

In the submitted manuscript, Sha and co-workers investigated the effects of supplementary feeding in the cold season on the rumen microbes, VFAs, and expression of SGLT1 in the rumen epithelium of cattle-yak. Overall, the topic of manuscript is of interest. This study applied 16S rRNA sequencing and covered the functional analysis of KEGG pathways and COG analyses. While the reviewer has major concerns regarding the experimental design/discussion part.

The English sentence structure should be improved in some sentences.

The figure legends should be modified, adding the comparision of treatment groups and statistical analysis (P-value) under the tables.

Experimental design

This study provided changes in microbiota population and VFA, however, no details about animal host (gender) and the power of statistic calculation for the number of animals in each treatment group were presented since these factors can react to these changes and statistical analysis. Also, the experimental design should be crossover designed with including the washout period to reduce the variation of animals, which reduced the readability and robust of this manuscript. Does the author have any explanation for the experimental design without crossover animals?

For microbial profiles, the family level is missing in the results. The authors represented the phylum and genus level in the abstract and results. The phylum consisted of a variety of microbial community, which have diverse function; therefore, the phylum is not well representative of the microbial community. It showed only general trend, the family and genus level are more validity. Please explain why you used this phylum level.

Regarding to the functional analysis, the results of KEGG pathways and cog genes from the experiment can be used to explain the changes of microbial population. Why does the author decided to put in the supplementary materials? I recommned to put in the main manuscript. Also, few results presented in the result and discussion section. I would recommend explaining more details related the functional analysis, not only mentioned “carbohydrate metabolism”

For concerning to the gene expression, Actually, there are other genes involving in glucose transporter found in the mucosa. What bases were these particular genes selected? Why did you interest only the expression of SGLT1?

Validity of the findings

In general, the finding provided the sufficient results according to the method. However, the data presentation should be improved. There are several repetitions in the results and discussion and the conclusions are not supported. It must be strengthened. All over-interpret data must be removed. Please reconsider and rephrase. The manuscript would gain substantially by making better use of the wealth of literature data for interpretation of the experimental results.

Additional comments

Abstract part:
Please reconsider to report the results at the family and genus level.
Introduction part:
Line 55-56: The author stated “some glucose in the rumen is transported directly to the blood via sodium ion-dependent glucose transporter 1 (SGLT1), thereby saving energy for fermentation (Aschenbach et al., 2000; Aschenbach et al., 2000; Aschenbach et al., 2002)”. The authors should provide clearly understanding for your text how it saves the energy for fermentation.
Line 74-79: The hypothesis is missing.
Materials and Method:
Line 84-85: Ingredient and chemical composition of experimental diets should be declared.
no animal gender related data was provided. Does the author use crossover design?
Line 86-88: How special is the “supplementary feeding “?
Please reconsider to explain factors “soluble or digestible or…………..”
Line 105: The condition for processed the library and sequenced on an Illumina MiSeq platform (Illumina, San Diego, CA, USA) should be provide.
Line 149: “The microbial communities in each group (identified by 16S rRNA sequencing), based on both the Kyoto Encyclopedia of Genes and Genomes (KEGG) and Clusters of Orthologous Groups (COG) databases”. Does the author report the relative abundance at what level?
Result part:
Line 168: Paragraph “Microbial community structure in the rumen”
Please reconsider to provide the result at the family level.
Does the author report the relative abundance at what level?
Line 200: The sentence is hard to follow.
Figure
Discussion part:
Overall: Please remove all repeated results “were/was significantly” “higher or lower” throughout the paragraph. Please provide more explanations and give more reason and scientific data from previous literatures or your correlating results from the experiments to support how the diet-related shift in microbiota and functional capability.

Line 210-215: “Firmicutes possesses many genes encoding energy metabolism-related enzymes”. The sentence is over-representing data. The Firmicutes comprised of several microbial group and may shift their functions depending on the substrate availability. Therefore, presenting the specific microbial groups at family and genus level would gain substantially by making better use of the wealth of literature data for interpretation. Also, the microbial group should be consistent with the KEGG findings on carbohydrate metabolism.
Line 223-226: Please add references
Line 240: “Therefore, we speculate that appropriate supplementary feeding after grazing in the cold season can promote the degradation of carbohydrates in the feed, which can allow the cattle-yak to more effectively cope with the harsh cold season”
The sentence is over-interpreting data. The authors need to explained how diet-related change in microbial groups and shifts in KEGG and COG pathways. Not only Carbohydrate metabolism, but also other pathways should be discussed.
Line 243-245: “In both the KEGG and COG analyses, the carbohydrate metabolism gene families”
Please provide the lists of gene families which affected by the diets.
Line 263-264: Please consider to interpret the data with the result of the KEGG and COG analyses
Line 274: Please add references

Conclusion part:
The conclusions are repeated from the result and not supported the finding. Provide more novel data further research.

---

## Round 0.2 · Minor Revisions

After the first revision, your article has been improved. However, our reviewer still had some comments that need you to revise a manuscript again. The important issue is English, we need you send this article for proof by a fluent speaker or Language institute. Please submit the document that article was English proofread with the next submission. I am looking forward to getting your revised manuscript.

·

Basic reporting

In general, the revised version is much better but the reviewer has major concerns regarding the grammar error and conclusion part. The Figures and Tables legends should be expand and also include the treatment groups in the titles.

Experimental design

No comment

Validity of the findings

In general, the data presentation has improved from the previous version. The discussion and conclusion must be strengthened and removed all repetation.

Additional comments

Materials and Method:
Line 94: Typing error “…..”
Line 100 - 105: Grammar error and wrong sentence structures. The sentence in the Materials and Method part should be past tense. Also, these are not sentence “Take 50ml of the rumen contents for each cattle-yak and filter the rumen content with four layers of sterile gauze. Divide the filtered rumen fluid into 3 sterile cryo tubes, which were immediately placed in liquid nitrogen for preservation, and bring them back to the laboratory for further processing 16S rRNA analysis”
Please recheck the whole Materials and Method part.
Line 128: Please rewrite the sentence. “The detector temperature was “………” Better to combine with previous sentence.


Line 144: it should have verb after and
“and ……………………….chimera removal (UCHIME v4.2) in order to obtain optimized sequences (tags)”

Line 149: Please reconsider to change the sentence. “P<0.05 is significant difference, p<0.01
168 is extremely significant difference”. The extremely significant” is not scientific. Better to use “considered significant levels at ………..”
Result part:
Line 177: grammar error. It should be past tense. “ indicate the species diversity and species richness”
Line 204-211: grammar error. It should be past tense. Group G+S is significantly higher than group G. Among the 25 COG gene families, transcription was the most frequent. In addition, the most abundant functional Carbohydrate transport and metabolism, Amino acid transport and metabolism, and Energy productions and conversion are significantly different. Among, Carbohydrate Transport and Amino acid transport and metabolism show as G group with significantly high G+S group. However, in terms of Energy Productions and Conversion, G group is significantly lower than G+S group, which is consistent with the KEGG function analysis above.

Line 246-249: grammar error. It should be past tense
Discussion part:
Overall: Please remove all repeated results “were/was significantly” “higher or lower” throughout the paragraph. You can keep the same meaning but paraphrase the sentences.
Line 260: grammar error. It should be past tense
Line 305-310: grammar error. It should be past tens

Conclusion part:
The conclusions are repeated from the result and very weak. Please rewrite the whole section. Provide more novel data further research.
The structure of conclusion should be
1 Restate your research topic.
2 Restate the thesis.
3 Summarize the main points.
4 State the significance or results.
5 Conclude your thoughts./ Provide next steps

---

## Round 0.3 · accepted · Accept

Congratulations on your publication.